



# Estimating optical extinction of liquid water clouds in the cloud base region

Karolina Sarna[1], David P. Donovan[2], and Herman W.J. Russchenberg[1]

[1]TU Delft Climate Institute, Faculty of Civil Engineering and Geotechnology, Delft University of Technology, Stevinweg 1, 2628 CN, Delft, The Netherlands
[2]Royal Netherlands Meteorological Institute (KNMI), P.O. Box 201, 3730AE, De Bilt, The Netherlands

*Correspondence to:* Karolina Sarna (k.sarna@tudelft.nl)

**Abstract.** Accurate lidar-based measurements of cloud optical extinction, even though perhaps limited to the cloud base region, are useful. Arguably, more advanced lidar techniques (e.g. Raman) should be applied for this purpose. However, simpler polarization and backscatter lidars offer a number of practical advantages (e.g. better resolution, more continuous and numerous time series). In this paper we present a backscatter lidar signal inversion method for the retrieval of the cloud optical extinction

5    in the cloud base region. Though a numerically stable method for inverting lidar signals using a far-end boundary value solution has been earlier demonstrated and may be considered well-established (i.e. the Klett inversion), the application to high-extinction clouds remains problematic. This is due to the inhomogeneous nature of real clouds, the finite range-resolution of many practical lidar systems and multiple-scattering effects. We use an inversion scheme where a backscatter lidar signal is inverted based on the estimated value of cloud extinction at the far end of the cloud and apply a correction for multiple-

10    scattering within the cloud and a range resolution correction. By applying our technique to the inversion of synthetic lidar data, we show that for a retrieval of up to 90 m from the cloud base it is possible to obtain the cloud optical extinction within the cloud with an error better than 5%. In relative terms, the accuracy of the method is smaller at the cloud base but improves with the range within the cloud until 45 m and deteriorates slightly until reaching 90 m from the cloud base.



# 1 Introduction

Lidar was used to probe the atmosphere ever since 1960 (e.g., Collis, 1966; Fiocco and Smullin, 1963). Lidar measurements facilitate characterisation of the atmosphere and have many different applications, including determining properties of aerosols (Müller et al., 1998) and clouds (Turner, 2005).

Lidars possess a unique ability to observe the optical properties of clouds such as cloud extinction coefficient ($\alpha$). Through an inversion of the backscattered power received by a lidar system, an estimate of the cloud extinction coefficient can be retrieved (Klett, 1981). This optical property of the cloud can be linked to cloud's microphysical properties (Kokhanovsky, 2004). Although lidar can only penetrate a small part of a cloud, typically 100 to 300 meters from the cloud base, the cloud base region is of a strong interest for studies concerned with cloud formation and aerosol-cloud interactions (McComiskey and

Feingold, 2012).

Despite the long history of lidar measurements and the vast amount of data available, very few quantitative evaluations of the cloud optical extinction retrieval accuracy under realistic conditions exist (e.g., Carnuth and Reiter, 1986; Rocadenbosch et al., 1998). Lidar signal inversion in realistic conditions is more difficult due to the effects of finite lidar range resolution and multiple-scattering occurring within the cloud.

In this paper we present a procedure to retrieve the cloud optical extinction coefficient, using a single field of view (FOV) depolarization lidar. We use the Klett solution (Klett, 1981) with the inclusion of a multiple-scattering correction (Hu et al., 2006; Roy and Cao, 2010) and an explicit treatment of the molecular and cloud contributions to the returned signal (Fernald, 1984). We demonstrate, using synthetic lidar signals generated using a Monte-Carlo RT model fed with Large-Eddy simulation (LES) fields, that useful extinction profiles can be retrieved using simple elastic polarization lidars.

The outline of the paper is as follows. In Sect. 2 we present background material. In Sect. 3 we give a brief description of the EarthCARE Simulator (ECSIM) and scenes created for this investigation. Sect. 4 presents the results of the inversion and discusses the issues related to conducting accurate inversions and present our methodology to address them. We conclude the paper with a summary of the findings and an outlook of possible applications.

# 2 Lidar signal inversion

The single-scattering lidar equation for a two-components atmosphere (cloud and molecular) can be defined as

$$P(z) = \frac{C_{lid}}{z^2} \left( \beta_{c,\pi}(z) + \beta_{m,\pi}(z) \right) e^{-2 \int_0^z \left( \alpha_c(z') + \alpha_m(z') \right) dz'}, \tag{1}$$

where $z$ is the altitude, $P(z)$ is the received power as a function of altitude, $C_{lid}$ is the lidar calibration constant, $\beta_\pi$ is the atmospheric backscatter coefficient, $\alpha$ is the atmospheric extinction coefficient and the 'c' and 'm' subscripts distinguish between cloud and molecular backscatter and extinction Fernald (1984). As the Klett solution applies strictly to a one-component at-

30 mosphere we introduce $\alpha'$ and $P'$ in order to account for the mixed contributions from cloud/aerosol and molecular scattering Fernald (1984). If we define

$$\alpha'(z) = \alpha_c(z) + S(z)\beta_{m,\pi}(z), \tag{2}$$



and

$$P'(z) = SP(z)e^{2\int_0^z \left(\alpha'_m(z')\right)dz'}e^{-2\int_0^z \left(S\beta_m(z')\right)dz'}. \tag{3}$$

Then Equation 1 can be recast as

$$P'(z) = \frac{C_{lid}}{\frac{S(z)}{z^2}}\alpha'(z)e^{\left(-2\int_o^z \alpha'(z')dz'\right)}, \tag{4}$$

which has the general form of the single-component lidar equation and has the well-know solution.

In order to calculate the optical cloud extinction coefficient, $\alpha'$, we invert Equation 1 following the analytical solution to the lidar equation proposed by Klett (1981).

$$\alpha'(z) = \frac{\frac{P'(z)z^2}{P'(z_0)z_0^2}}{\frac{1}{\alpha'_0} + 2\int_z^{z_0}\left(\frac{P'(z)z^2}{P'(z_0)z_0^2}\right)dz'}, \tag{5}$$

where:

$$\alpha'_0(z_0) = \alpha_c(z_0) + S\beta_{m,\pi}(z_0). \tag{6}$$

$S$ is the extinction-to-backscatter ratio ($S = \alpha(z)/\beta_\pi(z)$ here assumed to be range independent within the cloud) and for the water clouds and wavelengths in the range from 200 to 1064 nm it is around 16 sr (Yorks et al., 2011). $\alpha_0$ is the extinction coefficient at a reference height $z_0$. Following the method established by Klett (1981) and later Fernald (1984) we estimate the value of the extinction coefficient at the far end of the range interval to retrieve the full profile of the extinction coefficient.

This method was tested for cloudy and foggy conditions and proved appropriate for retrieving the extinction values and it shows small dependence on the estimated extinction boundary value ($\alpha'_0$) when the optical thickness of the range interval is increasing (Klett, 1985; Carnuth and Reiter, 1986).

Although the principle of this method of lidar signal inversion is straightforward, there is a number of issues that must be addressed to ensure accurate results. Section 4.1 outlines these difficulties and presents possible ways of overcoming them. In

this work we make use of simulated lidar signals for which the 'true' extinction profiles are know. The simulations include the effects of realistic cloud structure, the effects of finite lidar range resolution and lidar multiple-scattering.

## 3   ECSIM Simulations

To evaluate the retrieval of the cloud extinction we use synthetic signals produced using the lidar Monte-Carlo radiative transfer model component of the EarthCARE simulator (ECSIM) which has been modified for ground-based simulations (Donovan

et al., 2015). ECSIM is a tool to simulate measurements of four instruments, namely: the 94-GHz cloud profiling radar, the high spectral resolution lidar at 353 nm, the multi-spectral imager and the broad-band radiometer. The lidar model takes into account polarization, multiple-scattering and the effects of finite lidar range resolution. The ECSIM radar model was also used in this paper in an ancillary role. To retrieve information about the cloud extinction we only need information from lidar.





However, information from radar can be used for a further analysis of the scene. Radar can add the capability to identify regions of drizzle. It can also penetrate through a liquid water cloud and hence is useful for establishing the height of the cloud top.

To create the scene used in this work, a liquid water content (LWC) field was generated by a Large Eddy Simulation (LES) and introduced to ECSIM. The LES case used was corresponding to one from the FIRE campaign (Albrecht et al., 1988).

The ECSIM simulation used specifically an output from the Dutch Atmospheric LES model (DALES) (Heus et al., 2010). DALES utilizes a two-moment bulk scheme to model precipitation (Khairoutdinov and Kogan, 2000), where condensed water is qualified as either cloud water or precipitation and the number density of cloud droplets is prescribed. The ECSIM scene is created based on a snapshot of parameters extracted from DALES. Those parameters include temperature, pressure, non-precipitable cloud water, precipitation water content and precipitation droplet number density. Further, an explicit droplet size

distribution (DSD) is needed to create an ECSIM scene. As DALES does not provide DSDs, imposed DSDs were used, based on the DALES output. The precipitation mode DSD was based on the one from Khairoutdinov and Kogan (2000). The cloud mode DSDs were found by assuming modified gamma type distribution with a width parameter of 5 and assuming a constant cloud-number density, the effective radius of the distributions was then calculated using the model LWC fields.

Figure 1 presents the cross section of the Radar Reflectivity Factor and the Attenuated Backscatter Coefficient of the used cloud

scene. For this study we performed two simulations based on the same DALES output. One of the cloud scenes was made to simulate Attenuated Backscatter Coefficient with the inclusion of multiple-scattering effects (refereed to later in the text as $B_{MS}$) and the second simulation was made for the single scattering Attenuated Backscatter Coefficient (refereed to later in the text as $B_{SS}$). This allowed us to directly compare the impact of the multiple-scattering on the retrieved values of the extinction coefficient, as well as evaluate the correction for the multiple-scattering presented in Section 4.1.2.

## 4   Inversion results

### 4.1   Difficulties in inversion steps

#### 4.1.1   Defining the normalization interval

In order to obtain a profile of the optical cloud extinction from lidar returns we need to invert the received power (Eq. 1) into a cloud optical extinction coefficient as explained in Sec. 2. Following the solution proposed by Klett (1985) it is necessary to

define the range interval where the signal can be normalized. The value of extinction, $\alpha'_0$, is estimated at a certain height, $z_0$, based, on the slope of the least square straight line fitted to the curve $ATB = ATB(z)$. The value of $\alpha'_0$ is calculated as follows

$$\alpha'_0 = -\frac{1}{2}\frac{dlnATB}{dz}, \qquad (7)$$

where $ATB$ is the Attenuated Backscatter Coefficient ($ATB(z) = P(z)z^2$) and $dz$ is the range resolution. Figure 2 presents the profile of the cloud optical extinction retrieved based on the slope method. It shows clearly that the slope method is not accu-

rate at the cloud base and the retrieved values get closer to the true extinction only at a certain height within the cloud. This is in accordance with a proposition by (Klett, 1981), who postulated that the normalization height $z_0$ where the value of $\alpha'_0$ is





estimated should be located at the far end of the cloud.

Another important aspect in deciding on the height of the normalization interval is the profile of the Attenuated Backscatter Coefficient (*ATB*). In order to calculate $\alpha'_0$, *ATB* at the chosen height has to be still usable, meaning that the noise level cannot be too high. Figure 3 presents the signal profile with marked normalization interval. Note that the interval is above the peak

5 of the signal and just before signal starts to be noisy or lost. In this study we chose a threshold for the *ATB* usability in the normalization interval at Signal-to-Noise Ratio (*SNR*) of 20. We tested the sensitivity of the inversion method to different values of *SNR* and found that values below 20 tend to influence the retrieval in the higher parts of the cloud. The first four bins within the cloud (up to 60 m within the cloud) are only affected by a mean error increase of 3 %. If *SNR* is below 20 then the normalization interval has to be set at a lower height.

### 4.1.2 Correcting the multiple-scattering

Measurements of water clouds by lidar backscatter always involve some contribution from multiple-scattering. In this study we use the multiple-scattering correction based on the accumulated depolarization ratio ($\delta_{acc}$) introduced by Hu et al. (2006) and further demonstrated by Cao et al. (2009). Lidar multiple-scattering occurring in water clouds can be linked to the depolar-

15 ization ratio. At 180° backscatter direction single scattering of spherical droplets retains the polarization of the incident light. However, scattering at different scattering angels changes the polarization state. For the liquid water clouds the depolarization of the signal can be attributed to the multiple-scattering (Sassen and Petrilla, 1986).

Based on the above described characteristics of water clouds and lidar backscatter Hu et al. (2006) described a relation between the linear depolarization of the backscatter signal and the fraction of multiple-scattering present in that signal. Based on the

20 Monte Carlo simulations of the multiple-scattering signals for numerous scenarios and different fields-of-view they derived the following relation:

$$A_S(z) = \frac{I_S(z)}{I_T(z)} \approx \frac{(1 - \delta_{acc}(z))^2}{(1 + \delta_{acc}(z))^2}, \tag{8}$$

where $I_S(z)$ is the integrated range-corrected single scattering signal and $I_T(z)$ is the integrated, range-corrected total-scattering signal (single and multiple-scattering). Both signals are integrated between the cloud boundaries, where cloud base height is established based on the lidar measurements and we use the top of the normalization interval instead of the cloud top as measurements above that height are no longer relevant. $\delta_{acc}(z)$ is the accumulated depolarization ratio. It can be calculated from the parallel and perpendicular components of the total backscattering signal:

$$\delta_{acc}(z) = \frac{I_{T,\perp}(z)}{I_{T,\parallel}(z)}, \tag{9}$$

where $I_{T,\perp}(z)$ is the total integrated perpendicular backscattered signal and $I_{T,\parallel}(z)$ is the total integrated parallel backscattered

signal.

In order to calculate the signal corrected for the multiple signal, in other words the signal contributed only to the single



scattering $ATB_{SS}$, we use the following formula:

$$ATB_{SS}(z) = A_S(z)ATB_{MS}(z) + I_T(z)\frac{dA_S}{dz}, \qquad (10)$$

where $A_S$ is the correction factor calculated from Eq. 8, $ATB_{MS}$ is the total range corrected signal, the $I_T(z)$ is the integrated, range-corrected total-scattering signal and $\frac{dA_S}{dz}$ is the derivative of the correction factor from Eq. 8. The last term of Eq. 10 can

be used to evaluate the depolarization both in simulated and real conditions. The value of $\frac{dA_S}{dz}$ should always be negative within the cloud because higher within the cloud more multiple-scattering occurs and a smaller part of the signal can be associated only with the single scattering.

Figure 4 presents samples of retrieved profiles with and without the correction for the multiple-scattering (noted as *MS* correction) plotted against the cloud optical thickness ($\tau$). Applying the MS correction improves greatly the accuracy and minimizes

the error of the retrieved profiles (for more detailed information see Table 2). Based on the data analysis performed for this paper we can conclude that multiple-scattering correction has a big impact on the accuracy of the retrieved cloud optical extinction.

### 4.1.3 Effects of the range resolution

The finite range resolution of the lidar signal is another factor that influences the final results of the inversion. The range

resolution of lidar varies depending on the system and the larger it is the higher might be its impact on the final inversion results. Problems with the resolution of lidar were mentioned before (Evans, 1984), but were never really studied and no solution to the problem was proposed so far.

The difficulty associated with the range resolution occurs since practical lidar data is always acquired at a finite resolution and thus must be interpreted using a discrete form of solution to the lidar equation. The continuous form of the equation 5 is often

naively transformed into a discrete form, where the integration is transformed into a summation using e.g. the trapezoid rule, yielding

$$\alpha_i' = \frac{\frac{P_i' z_i^2}{P_{i_0}' z_{i_0}^2}}{\frac{1}{\alpha_0'} + P_i' z_i^2 \Delta z + 2\sum_{i+1}^{i_0-1} P_i' z_i^2 \Delta z + P_{i_0}' z_{i_0}^2 \Delta z}. \qquad (11)$$

Although this is a common practice when transforming continuous equation to discrete form in algorithms, it may not be sufficiently accurate. If the value of $\alpha' \Delta z$ is small enough, then the approximation by the use of the trapezoid rule is accurate

and the resulting value of $\alpha'$ corresponds to the bin mid-point. However if that value is large, the applied approximation is not correct anymore. The detailed explanation of the calculations is presented in A.

Based on the calculations for the mid-point of the bin we define the resolution correction (*RES* and *RES$_2$*) as follows:

$$RES(z) = \frac{e^{\alpha'(z)\Delta z}}{e^{\alpha'(z)\Delta z} - e^{-\alpha'(z)\Delta z}}, \qquad (12)$$

and

30 $$RES_2(z) = \frac{2\alpha'(z)\Delta z}{e^{\alpha'(z)\Delta z} - e^{-\alpha'(z)\Delta z}}, \qquad (13)$$


where $\alpha'(z)$ is the retrieved cloud optical extinction and $\Delta z$ is the height resolution.

In order to apply this correction factor we need to perform the inversion in two steps. Firstly, we invert the lidar signal and apply the multiple-scattering correction. The resulting optical cloud extinction ($\alpha$) from the first inversion is used in the range resolution correction (Eq. 12 and 13) and then the corrected signal is inverted again.

Figure 4 presents the retrieved profiles of $\alpha$ with the multiple-scattering correction (denoted as *MS*) and with the multiple-scattering correction together with the range resolution correction (denoted as *MS & RES*). We observe that while the *MS* correction on its own improves the retrieval greatly, after application of the *RES* correction values of $\alpha$ are closer to the true value of extinction coefficient. The importance of the resolution correction can be easily presented when we inverted the simulated single scattering signal ($B_{SS}$, as mentioned in Section 3). Table 2 presents error and accuracy of the inversion results
(as described in Section 4.3).

## 4.2 Estimating cloud base height

Although it is not directly connected to the inversion procedure, an accurate estimation of the cloud base height is also a challenging problem in cloud observation. In this study we use the peak of the lidar perpendicular signal to evaluate the cloud base height. Lidar power ($P(z)$, Eq.1) from a depolarization lidar can be divided into the parallel ($P(z)_{\parallel}$) and perpendicular power
($P(z)_{\perp}$). In every profile we find the peak of the perpendicular power ($P(z)_{\perp\ max}$) and estimate the cloud base to be at the height where $P(z)$ is equal or greater than $P(z)_{\perp\ max}$ divided by ten. We found that this estimate predicts the height of cloud base with a good accuracy for the liquid water clouds. Figure 1 presents the Radar Reflectivity Factor and the Attenuated Backscatter Coefficient for the scene used in this study. Both panels present the estimate of the cloud base height marked with a magenta line. Examining the panel with the *ATB* we see that our estimate is a good approximation.

## 4.3 Signal inversion error and accuracy

In this study we use the ECSIM cloud scene to test the accuracy and estimate the error of the lidar signal inversion. The dataset from ESCIM gives us information about the true value of optical extinction coefficient within the cloud. Thanks to that we can calculate the percent error and the accuracy of the inversion method by comparing the retrieved value to the true (simulated)
value of the optical extinction coefficient. For those calculations we use the following formulas:

$$E_{B_{SS}or B_{MS}} = \frac{\alpha_{retrieved} - \alpha_{simulated}}{\alpha_{simulated}} * 100\%, \tag{14}$$

to estimate the percent error, and:

$$A_{B_{SS}or B_{MS}} = \frac{\alpha_{retrieved}}{\alpha_{simulated}} * 100\%, \tag{15}$$

to estimate the accuracy, where the subscript $B_{SS}$ is used when we are inverting signal from the single scattering simulation
and the subscript $B_{MS}$ is used for the simulations from the multiple-scattering simulations. For the whole dataset the mean values for each height above the cloud base are presented in Table 1 for $B_{SS}$ and in Table 2 for $B_{MS}$.





As we indicated before, values retrieved at the cloud base (defined as being 0 m from the cloud base in Table 1 and 2) are the ones with the biggest percent error. This stems from the difficulty in the signal inversion at very small values of cloud optical extinction. We observe a great improvement of the accuracy of the inversion further within the cloud. We present values of the inversion error and accuracy for the retrieval without any correction and for the retrieval only with the resolution correction

($A_{B_{MS} \text{ with RES}}$ and $E_{B_{MS} \text{ with RES}}$), only with the multiple-scattering correction ($A_{B_{MS} \text{ with MS}}$ and $E_{B_{MS} \text{ with MS}}$) and with both the multiple-scattering and the resolution correction ($A_{B_{MS} \text{ with RES\&MS}}$ and $E_{B_{MS} \text{ with RES\&MS}}$).

For the results of the inversion of the $B_{SS}$ signal we tested how can the resolution correction improve the results of the retrieval. Table 1 presents the mean error and accuracy calculated at different levels within the cloud. We observed an increased impact of the resolution correction deeper within the cloud. At a distance 45 to 90 m from the cloud base the resolution correction

almost doubles the accuracy. This is mostly due to an increase in the value of cloud optical extinction ($\alpha$). As we explain in the Appendix A, the resolution correction is less relevant for small values of $\alpha$. Inversion of the signal with the simulated multiple-scattering ($B_{MS}$), and thus far more resembling actual measurements, is understandably less accurate. Table 1 presents mean error and accuracy of the retrieved cloud optical extinction for different heights above the cloud base. Inversion without any correction had a mean error ranging from 40% at cloud base to 26% in the cloud. We observed that with the resolution

correction only the error can be improved by up to 3%. The correction for the multiple-scattering has a much bigger impact, it improve the inversion error by around 35% at the cloud base and by 20% higher within the cloud. By combining the resolution and multiple-scattering correction the error of the inversion can be improved to between 6% at the cloud base and 3-4% within the cloud. We observed that the inversion is most accurate between 30 and 60 m within the cloud. Figure 5 presents the cross-section of the retrieval percent error of the cloud optical extinction for the inversion of simulated multiple-scattering signal

with the inclusion of the resolution and multiple-scattering correction. The increase of the error above 60 m from the cloud base mainly is due to an underestimation of the value of cloud optical extinction at the normalisation height ($\alpha_0$).

The accuracy of the retrieval is connected to the cloud optical thickness. Figure 6 presents scatter plots of the retrieved values of $\alpha$ with the multiple-scattering and range resolution correction plotted against the modelled ones. The data is divided by the value of the optical thickness, $\tau$, where

$$25 \quad \tau(z) = \int_0^h \alpha'(z)dz, \tag{16}$$

$\alpha$ is the cloud optical thickness and $h$ is cloud depth. Every panel includes an imposed red line which represents an equality between the modeled and retrieved values. We also used a colour scaling, where the color bar represents the value of cloud optical extinction at every point. The error (Eq. 14) and accuracy (Eq. 15) for each bin on the optical thickness is also presented. We observed that the inversion method works best for the values of $\tau$ between 0.6 and 1.05. The error for values of $\tau$ above 1.5

is higher and the retrieved cloud optical extinction is underestimated. The probable cause of this behaviour of the retrieval is the loss of a signal with the increase of the cloud optical thickness. For the optical thickness below 0.6 and further below 0.15 the important factor influencing the accuracy of the retrieval is the estimation of the cloud base region.

Figure 5 presents the cross section of the cloud optical thickness and the retrieval percent error. Here again we can clearly see





that the percent error is highest close to the cloud base, ranging between 8%-15%, and deeper within the cloud it rarely exceeds 7%. This means that when inverting the lidar signal it is important to carefully examine the first range above the cloud base.

## 4.4 Impact of $\alpha'_0$ estimation

Klett (1981) stated that the value of $\alpha'_0$ does not influence much the final results of the inversion. In our study we tested

this statement by performing inversion with the actual value of extinction at the normalisation height $z_0$ instead of the value calculated from the slope method (7). The results of this inversion are presented in Table 3. The error for the inversion with the multiple-scattering and resolution correction is improved by around 0.5%. The error improvement is more significant for the values retrieved above 60 m from the cloud base. This is due to the underestimation of the value of $\alpha'_0$ with the slope method (Figure 2). We also tested the accuracy of the calculated $\alpha'_0$ by comparing it to the actual value of $\alpha$ at the normalization height

$z_0$. The mean accuracy of $\alpha'_0$ for the whole data set was 95%, with the minimum accuracy of 89% and the maximum one of 112%.

## 5 Conclusions

In this paper we presented a method of lidar signal inversion for the retrieval of the cloud optical extinction in the cloud base region. This method was first presented by Klett (1981). We showed that with the correction for the multiple-scattering within

the cloud and the resolution correction this method can be successfully used for the retrieval of the cloud optical extinction. Both those corrections are essential to improve the accuracy of the retrieved extinction profile and minimize the error. We presented the performance of the retrieval based on the synthetically created cloud scene where responses of the lidar to a specific cloud conditions were simulated. Even though in some case the cloud base was not varying much in height, the analyzed data indicated that signal inversion close to the cloud base (specifically at the range of the detected cloud base) is

prone to error. The retrieval of the cloud optical extinction works better at higher values of the optical thickness. It is therefore our recommendation to use only data points located at least one gate range above the detected cloud base height. We also showed that the approximation of $\alpha'_0$ calculated with the slope method can be used as an estimation of actual cloud optical extinction at the normalization height. More importantly, improving the value of $\alpha'_0$ by using the actual extinction at the normalization height does not improve the retrieved values significantly if the correction for the multiple-scattering and range

resolution is implemented.

We showed that the inversion of the lidar signal with the proposed corrections yields a good estimate of the cloud extinction. Not only is this method fast, but also, because of the use of a standard backscatter depolarization lidar, can be applied to multiple systems and used operationally. Through a link between cloud microphysical properties and the optical extinction this can provide a valuable dataset to be used in the studies of cloud microphysics and impacts of clouds on the climate.





## Appendix A: Derivation of the resolution correction

The difficulty associated with the range resolution occurs since practical lidar data is always acquired at a finite resolution and thus must be interpreted using a discrete form of the lidar equation. The single-scattering lidar continuous equation, in term of the range corrected signal $B(z)$ can be defined as:

$$B(z) = C\alpha'(z)e^{-2\int_0^z \alpha(z')dz'}, \tag{A1}$$

where $C$ is the lidar constant, $\alpha'$ is the cloud optical extinction and $z$ is range or in therms of optical thickness $\tau$ as:

$$B(z) = C\frac{d\tau}{dz}e^{-2\tau(z)}, \tag{A2}$$

where $\tau$ is the cloud optical thickness. In the discrete form, backscatter signal for one point $B_i$ is defined as

$$B_i = \int_{z_i-\frac{\Delta z}{2}}^{z_i+\frac{\Delta z}{2}} B(z)dz. \tag{A3}$$

Applying the form from the Eq. A2 we can say that:

$$B_i = -\frac{c}{2}e^{-2\tau(z)}\Big|_{z_i-\frac{\Delta z}{2}}^{z_i+\frac{\Delta z}{2}}, \tag{A4}$$

which is equal to

$$B_i = \frac{c}{2}\left[e^{-2\tau(z_i)} - e^{-2\tau(z_i+\frac{\Delta z}{2})} + e^{-2\tau(z_i-\frac{\Delta z}{2})} - e^{-2\tau(z_i)}\right] \tag{A5}$$

and

$$B_i = \frac{c}{2}[B_{i,1} + B_{i,2},] \tag{A6}$$

as ilustrated on figure 7. The difference between $B_i$ and $B_{i,1}$ can be then calculated

$$\frac{B_i}{B_{i,1}} = -\left(\frac{1 - e^{-2(\tau(z+\frac{\Delta z}{2})-\tau(z))}}{1 - e^{-2(\tau(z-\frac{\Delta z}{2})-\tau(z))}}\right) + 1. \tag{A7}$$

If we assume that

$$\tau(z+\frac{\Delta z}{2}) + \tau(z) \approx \frac{\alpha'\Delta z}{2} \tag{A8}$$

and

$$\tau(z+\frac{\Delta z}{2}) - \tau(z) \approx -\frac{\alpha'\Delta z}{2} \tag{A9}$$

Eq A7 becomes

$$\frac{B_i}{B_{i,1}} = -1\left(\frac{1 - e^{-\alpha'\Delta z}}{1 - e^{\alpha'\Delta z}}\right) = -\frac{e^{\alpha\Delta z} - e^{-\alpha'\Delta z}}{1 - e^{\alpha'\Delta z}}. \tag{A10}$$





We can then calculate $B_{i,1}$,

$$B_{i,1} = B_i \left( \frac{e^{\alpha' \Delta z}}{e^{\alpha' \Delta z} - e^{-\alpha' \Delta z}} \right) \tag{A11}$$

and thus we define the resolution correction *RES* which equals

$$RES = \frac{e^{\alpha' \Delta z}}{e^{\alpha' \Delta z} - e^{-\alpha' \Delta z}}. \tag{A12}$$

In cases when $\alpha' \Delta z$ will be large:

$$B_{i,1} \approx B_i \left( \frac{e^{\alpha' \Delta z}}{\alpha' \Delta z} \right) \approx B_i, \tag{A13}$$

and if $\alpha' \Delta z$ will be small:

$$B_{i,1} \approx B_i \left( \frac{\alpha' \Delta z}{2\alpha' \Delta z} \right) \approx \frac{1}{2} B_i. \tag{A14}$$

The value of *RES* will be around 0.5 and it's applied to the lidar power signal, specifically in the calculation of the integral
in the term $\int_z^{z_0} \left( \frac{P'(z) z^2}{P'(z_0) z_0^2} \right) dz'$ in Eq. 5, where the usual value of $\frac{1}{2}$ used in the trapezoidal rule of integration is replaced by a corresponding *RES*. If that values of *RES* will be higher or smaller we have to compensate so that the equalities of Eq. A, specifically $[B_{i,1} + B_{i,2}]$, are not greater than one. For that we derived the second part of the resolution correction $RES_2$. $RES_2$ is defined as

$$RES_2 = \frac{2\alpha' \Delta z}{e^{\alpha' \Delta z} - e^{-\alpha' \Delta z}}. \tag{A15}$$

The value of $RES_2$ cannot be higher than 1. $RES_2$ is applied in the first term of Eq. 5, so that $\frac{P'(z) z^2}{P'(z_0) z_0^2}$ becomes $\frac{(P'(z) z^2) RES_2(z)}{(P'(z_0) z_0^2) RES_2(z_0)}$.





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





**Table 1.** Mean error and accuracy of the cloud optical thickness extinction retrieval for different heights above the cloud base. Data is retrieved by inverting simulated single scattering signal ($B_{SS}$) signal with $\alpha'_0$ estimate calculated from Eq. 7. Results from two inversions are presented: one without any correction and one with the application of the resolution correction calculated from Eq. 12 and 13 (noted with the subscript *RES* )

| Distance from cloud base | $A_{B_{SS}}$ | $E_{B_{SS}}$ | $A_{B_{SS} \text{ with RES}}$ | $E_{B_{SS} \text{ with RES}}$ |
|---|---|---|---|---|
| 0.0 | 92.67% | 8.72% | 93.21% | 8.28% |
| 15.0 | 92.04% | 8.72% | 92.76% | 8.07% |
| 30.0 | 93.15% | 6.99% | 94.23% | 5.96% |
| 45.0 | 93.69% | 6.35% | 95.11% | 4.97% |
| 60.0 | 94.37% | 5.63% | 96.26% | 3.80% |
| 75.0 | 94.49% | 5.51% | 96.76% | 3.28% |
| 90.0 | 94.48% | 5.52% | 97.08% | 2.93% |

**Table 2.** Mean error and accuracy of the cloud optical thickness extinction retrieval for different heights above the cloud base. Data is retrieved by inverting simulated multiple-scattering signal ($B_{MS}$) signal with $\alpha_0$ estimate calculated from Eq. 7. Results from four inversions are presented: one without any correction, one with the application of the resolution correction calculated from Eq. 12 and 13 (noted with the subscript *RES*), one with the multiple-scattering correction calculated from Eq. 8 (noted with the subscript *MS*) and the last one with both the resolution and the multiple-scattering correction (noted with the subscript *RES&MS*)

| Distance from cloud base | $A_{B_{MS}}$ | $E_{B_{MS}}$ | $A_{B_{MS} \text{ with RES}}$ | $E_{B_{MS} \text{ with RES}}$ | $A_{B_{MS} \text{ with MS}}$ | $E_{B_{MS} \text{ with MS}}$ | $A_{B_{MS} \text{ with RES\&MS}}$ | $E_{B_{MS} \text{ with RES\&MS}}$ |
|---|---|---|---|---|---|---|---|---|
| 0.0 | 59.25% | 40.77% | 72.14% | 27.91% | 99.50% | 5.58% | 98.71% | 5.77% |
| 15.0 | 69.40% | 30.61% | 71.49% | 28.53% | 98.22% | 4.55% | 97.79% | 4.77% |
| 30.0 | 71.79% | 28.21% | 72.86% | 27.14% | 98.35% | 3.14% | 98.55% | 3.06% |
| 45.0 | 72.87% | 27.13% | 73.48% | 26.52% | 99.00% | 2.73% | 99.74% | 2.52% |
| 60.0 | 72.65% | 27.35% | 73.42% | 26.58% | 96.11% | 4.34% | 97.30% | 3.50% |
| 75.0 | 73.12% | 26.88% | 73.96% | 26.04% | 95.83% | 4.67% | 97.48% | 3.72% |
| 90.0 | 72.50% | 27.50% | 73.72% | 26.28% | 94.44% | 5.93% | 96.37% | 4.66% |





**Table 3.** Mean error and accuracy of the cloud optical thickness extinction retrieval for different heights above the cloud base. Data is retrieved by inverting simulated multiple-scattering signal ($B_{MS}$) with both the resolution and the multiple-scattering correction, with $\alpha_0$ equal to the true extinction at the normalisation height $z_0$ (noted as $\alpha_{true}$) and in the second case with $\alpha_0$ estimate calculated from Eq. 7 (noted as $\alpha_{slope}$).

| Distance from cloud base | $A_{B_{MS}}$ for $\alpha_{true}$ | $E_{B_{MS}}$ for $\alpha_{true}$ | $A_{B_{MS}}$ for $\alpha_{slope}$ | $E_{B_{MS}}$ for $\alpha_{slope}$ |
|---|---|---|---|---|
| 0.0 | 98.71% | 5.77% | 98.94% | 5.72% |
| 15.0 | 97.79% | 4.77% | 98.03% | 4.69% |
| 30.0 | 98.55% | 3.06% | 98.94% | 2.98% |
| 45.0 | 99.74% | 2.52% | 100.27% | 2.47% |
| 60.0 | 97.30% | 3.50% | 98.20% | 2.97% |
| 75.0 | 97.48% | 3.72% | 98.84% | 2.92% |
| 90.0 | 96.37% | 4.66% | 98.12% | 3.24% |

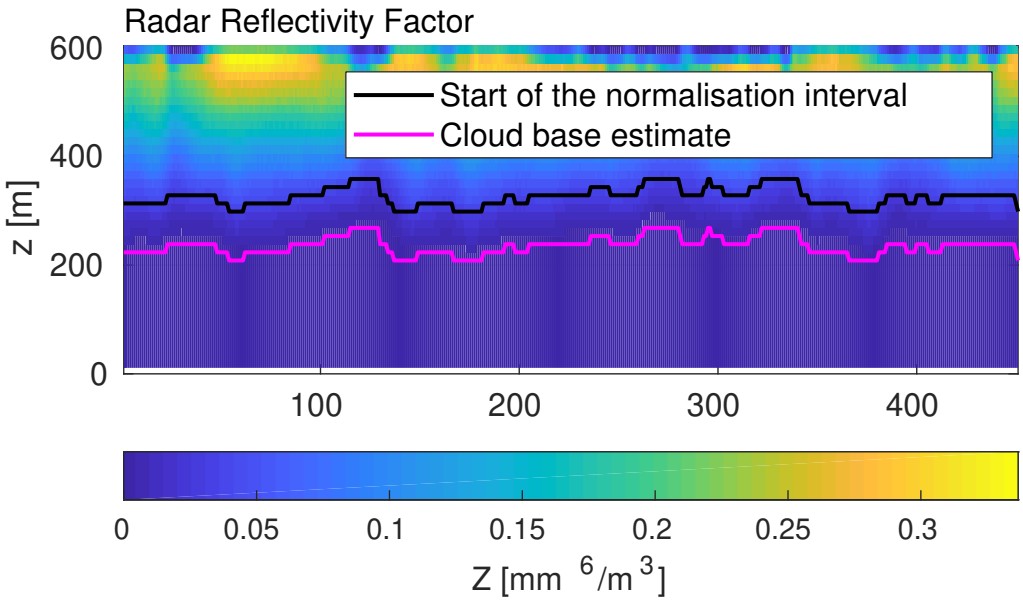

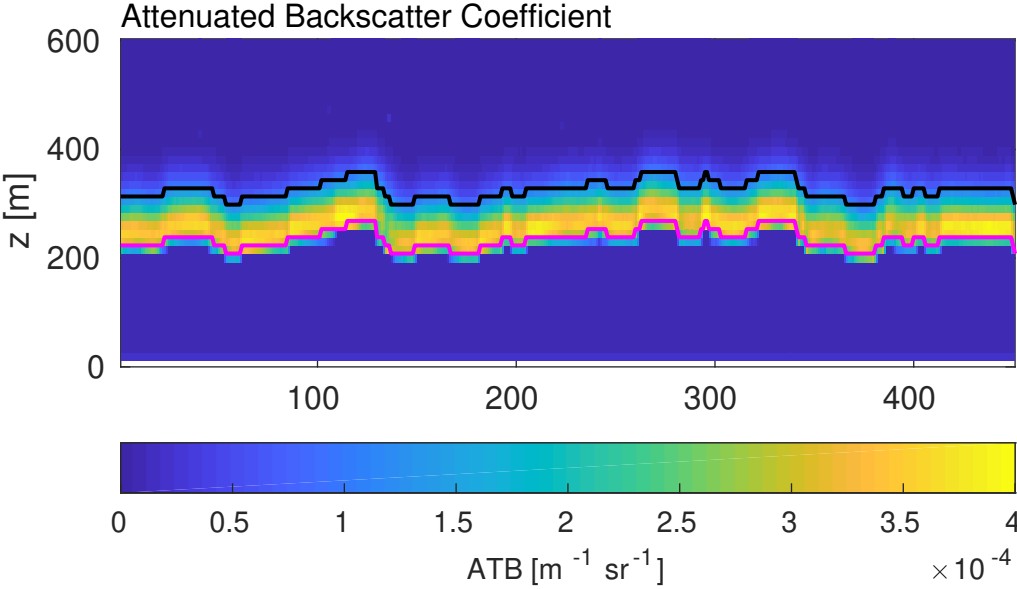

**Figure 1.** Cross section of the Radar Reflectivity Factor (top panel) and Attenuated Backscatter Coefficient (bottom panel) of the cloud scene produced with the ECSIM simulator. The magenta line indicates the estimate of the cloud base height and the black line indicated the beginning of the normalization interval.





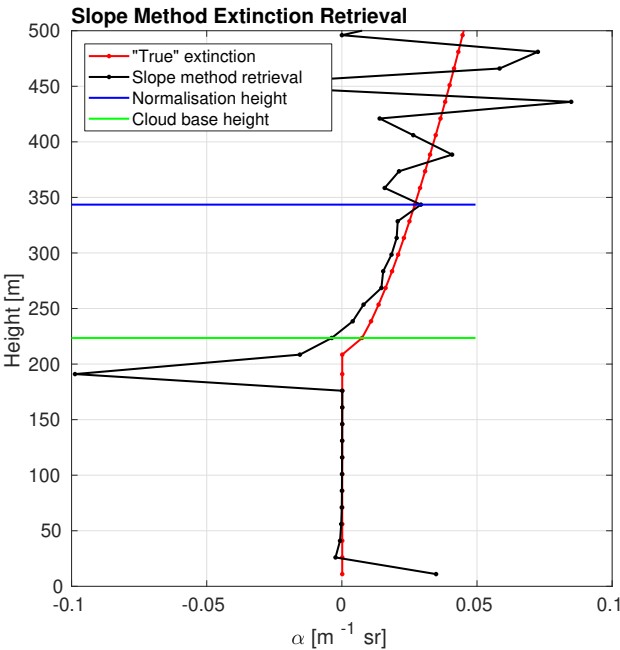

**Figure 2.** Profile of the extinction coefficient retrieved based on the slope method (Eq. 7) and the true extinction profile calculated from ECSIM.

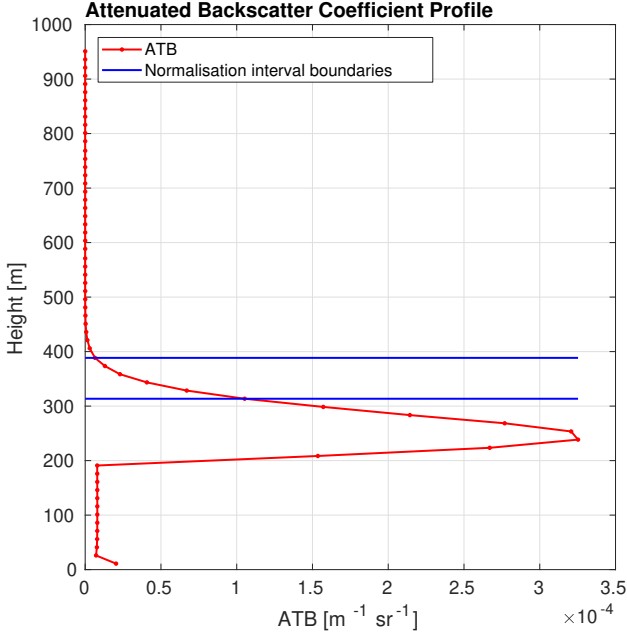

**Figure 3.** Profile of the Attenuated Backscatter Coefficient and boundaries of the normalization interval.



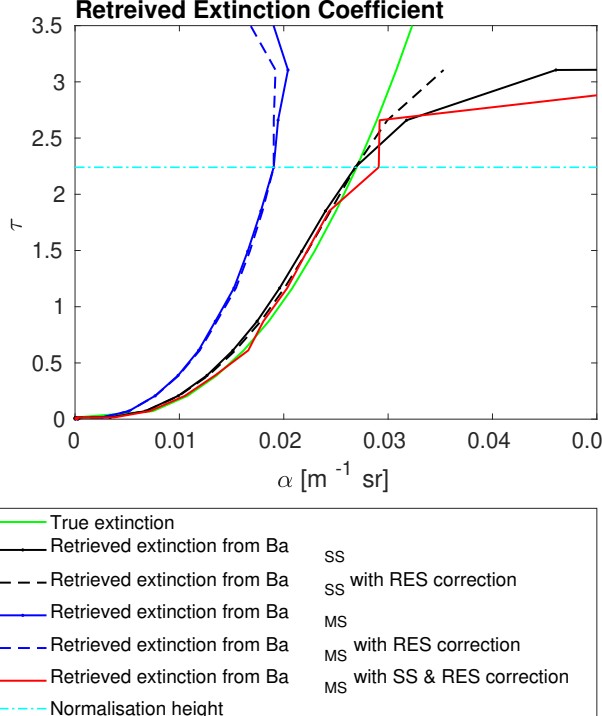

**Figure 4.** Profiles of the retrieved cloud optical extinction retrieved through an inversion of the signal with different corrections. The green line represents the true extinction calculated with the ECSIM. The black solid line represents the extinction profile retrieved without any corrections from the modeled single scattering attenuated backscatter. The dashed black line represents the extinction profile retrieved from the modeled single scattering attenuated backscatter with the resolution correction. The blue solid line represents the extinction profile retrieved without any corrections from the modeled multiple-scattering attenuated backscatter. The dashed blue line represents the extinction profile retrieved from the modeled multiple-scattering attenuated backscatter with the resolution correction. The red line represents the extinction profile retrieved from the modeled multiple-scattering attenuated backscatter with the resolution and multiple-scattering correction. The dashed cyan line indicates the beginning of the normalization interval.



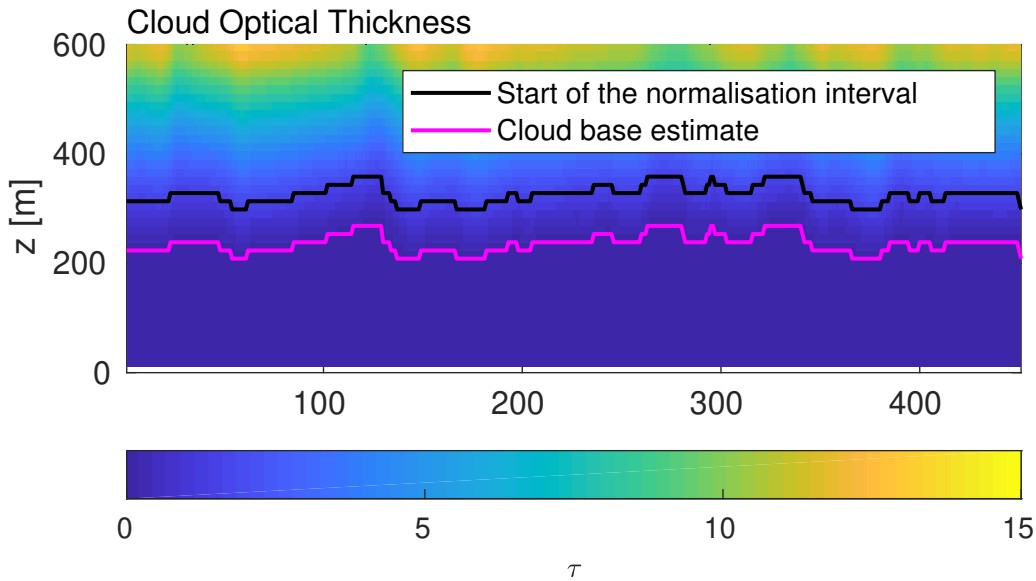

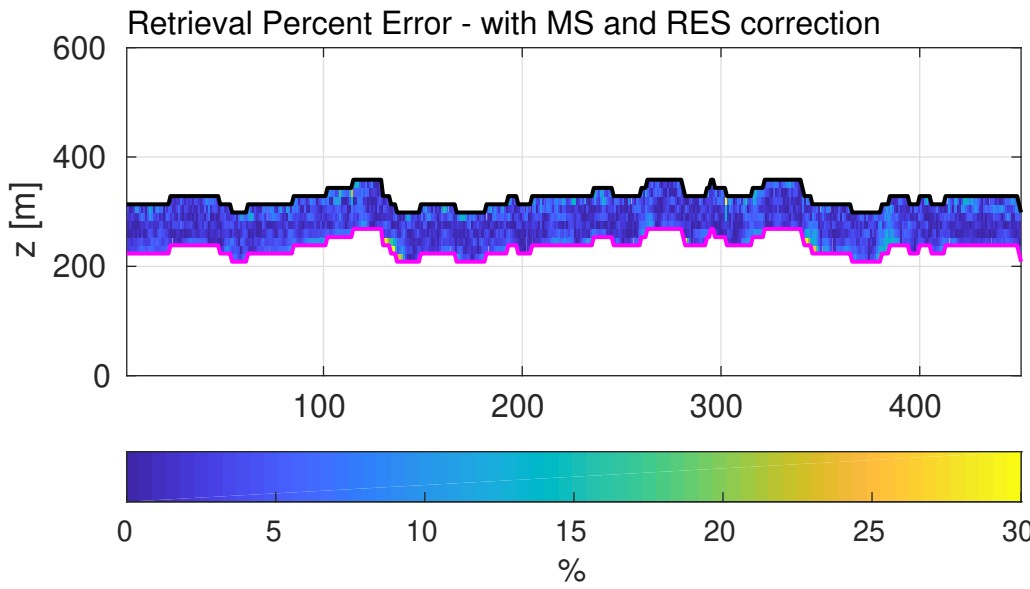

**Figure 5.** Cross section of the Cloud Optical Thickness (top panel) and Retrieval Percent Error of the cloud optical extinction retrieved with the multiple-scattering and range resolution correction (bottom panel). The magenta line on both panels represents the estimated height of the cloud base and the black line is the beginning of the normalization interval.



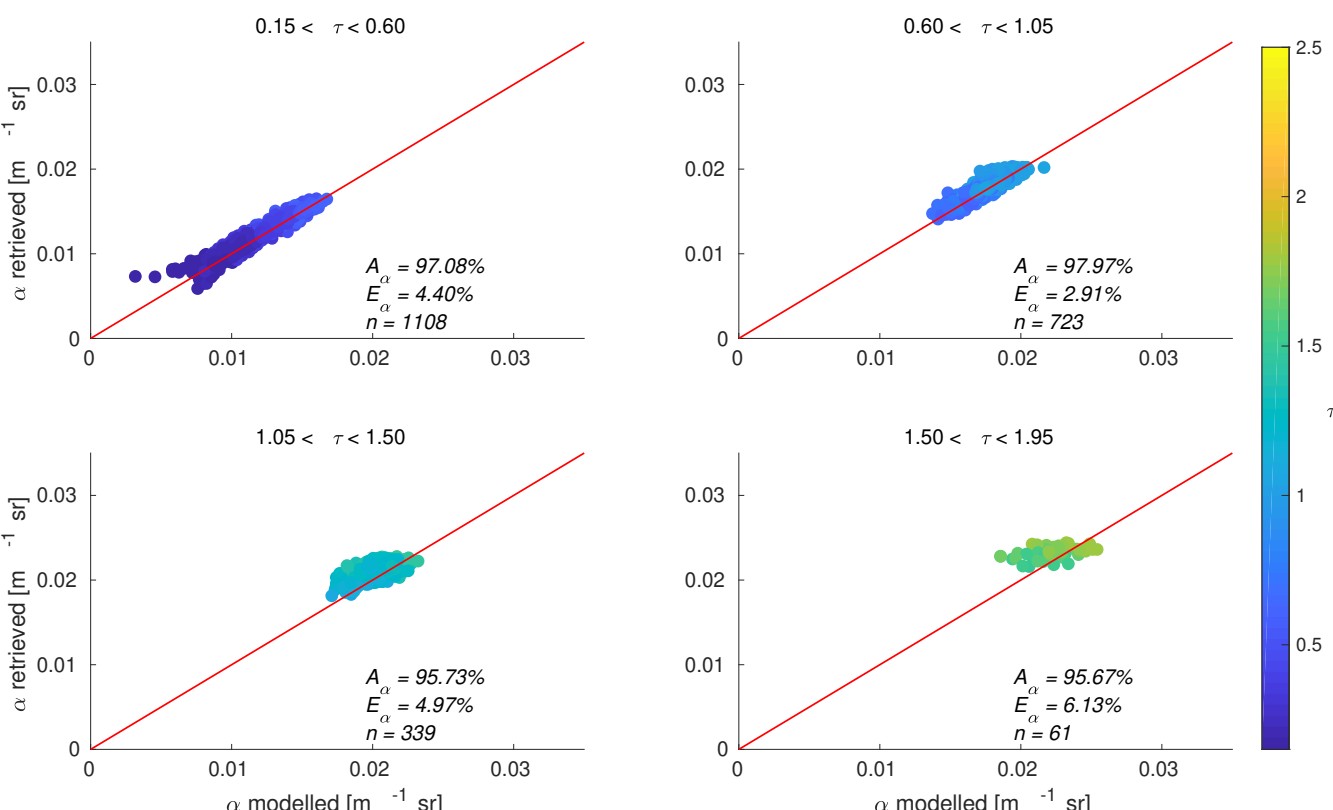

**Figure 6.** Scatter plots of the retrieved cloud optical extinction (with the multiple-scattering and range resolution correction) versus the modeled cloud optical extinction from the ECSIM divided into panels depending on the value of the optical thickness. The red line is imposed and represents the equality between the modeled and retrieved values. The color bar represents the value of the cloud optical thickness at each point. The error (Eq. 14) and accuracy (Eq. 15) for each bin of the optical thickness is also presented.

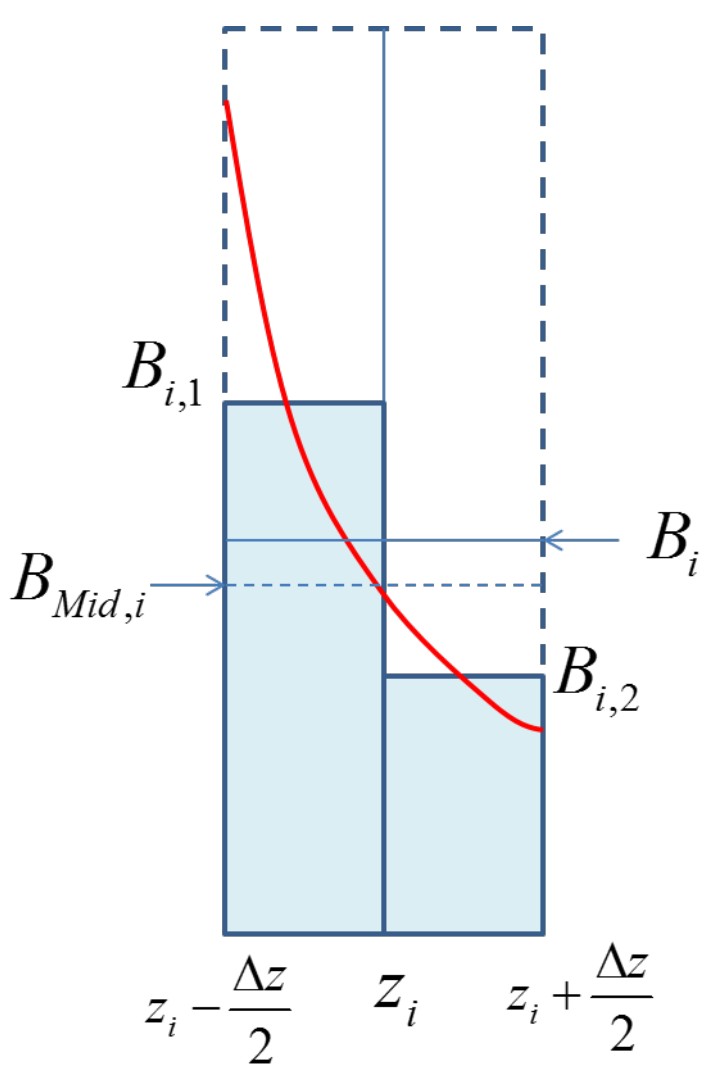

**Figure 7.** Illustration of the discrete form of the lidar equation.