# Peer review of "Estimating optical extinction of liquid water clouds in the cloud base region"

_Atmospheric Measurement Techniques, 2020_

## Author Response (AR1)

**Response to Anonymous Referee #1**

- *Only minor changes and a few typographical corrections are needed before the manuscript is ready for publication. Lidar characteristics (range resolution, field-of-view, etc.) from the EarthCARE simulator should be provided briefly in Section 3. Even if it matches what is presented in Donovan et al. (2015), a short overview is warranted here. One can gather from the tables that 15-m range resolution was used, but it should be explicitly stated.*

**Response:**

It is understood that the description needs to be reviewed. More detail on the dataset and on the specifics of the instruments used for the simulation will be provided in the reviewed version of the manuscript.

- *Secondarily, the use of accuracy, as defined in the manuscript, is somewhat misleading and almost redundant given the percent error is already provided. What connotation does the reader get from retrievals that are over 100% accurate? In such cases the percent error is the more meaningful measurement of the quality of your retrieval. To be fair, the ratio between the retrieved and simulated values has some usefulness. But in context of "accuracy", we get more from the percent error.*

**Response:**

This is a fair comment. The main idea was also to provide the deviation from the simulated values of the retrieved ones. This can be changed into a ratio in the revised version of the manuscript.

Specific Comments

- *Page 2, Line 8: Though it is stated in the title, distinguish that "lidar can penetrate only a small part of a cloud, typically 100 to 300 meters" refers specifically to liquid water clouds.*

**Response:**

Indeed, this will be specified explicitly in the revised version.

- *Page 5, Line 16: angels should be angles*

**Response:**

Will be corrected.

- *Page 6, Lines 8 – 12 or Page 7, Lines 5 – 10: In most cases, Figure 4 and Table 2 show the multiple scattering correction improves the extinction retrieval; however, from 75.0 –90.0 m or 1.8, the single scattering solution has a smaller error. Some comment/explanation to this point should be included.*

**Response:**

It should be noted that the single scattering solution is applied to a data that was simulated without the multiple scattering, hence indeed when the cloud optical thickness is higher (or simply with the increase of the altitude within the cloud) the solution has a smaller error as the contribution from the multiple scattering is not increasing as it would be in an actual cloud. The use of the simulations only with the single scattering is presented here to show that even if there were no multiple scattering occurring in the cloud, the resolution correction is still valid and can improve the retrieval. This will be explained better in the revised version of the manuscript to underline what are the differences between different simulated signals. Caption of Figure 4 will emphasize that both single-scattering and multiple scattering signal is simulated in

the corrected version of the manuscript.

- *Appendix Page 10, Line 6: therms should be terms*

**Response:**

Will be corrected.

- *Appendix Page 10, Line 16: Here difference is likely referring to the ratio, instead*

**Response:**

This will be corrected.

**Response to Anonymous Referee #2**

Specific comments:
- Equations 2-5:

*The presentation of Eqs.2–5 needs to be improved. It was hard to follow and check the derivations because of a few errors, such as alpha'_m in Eq. 3 should be jus alpha', and the term S/z^2 should be S*z^2 in Eq. 4, which according to my calculations would be redundant (or it is in Eq. 3).*

**Response:**

Indeed we agree that the equations 2 to 5 need to be improved. In Eq 3 alpha'_m should be just alpha_m and the extra fraction in eq 4 was removed. The changes will be submitted in the new version of the manuscript.

*It would be easier to follow, when the expression on page 3 line 11:S is the extinction-to-backscatter ratio (S = alpha (z)/beta_(z) here assumed to be range independent within the cloud) and for the water clouds and wavelengths in the range from 200 to 1064 nm it is around 16 sr (Yorks et al., 2011).... would be placed right before Eq. 2 and by adding to Eq. 2 =S (beta_c + beta_m).*

**Response:**

We agree with the comment, this will be adjusted in the new version of the manuscript.

*Finally, to my opinion, to obtain Eq.6, the apparent (i.e., multiple-scattering influenced) lidar ratio is needed in the Klett method (not the single-scattering lidar ratio, 18sr), and this quantity varies with multiple scattering impact and thus changes with height. Please clarify this, and state this clearly.... How did you overcome this effect?*

**Response:**

As explained in section 4.1.2 of the paper we are applying a correction for the multiple scattering (MS). This correction eliminates the need to use an effective S. We will underlie it in section 2 of the revised manuscript for clarity and provide reference to appropriate subsection.

- Section 3:

*Please provide more information of the computed scenes! Which form do the vertical profiles have? How many values of the extinction coefficient did you test?*
*Later on in Section 4.4, you report that the accuracy of ' for the whole data set was 95%. What is the data set?*

**Response:**
A more detailed description of the scene used will be provided in the revised version of the manuscript. The whole data set used for this study was showed on Figure 1. In total there were 450 profiles tested and an example of the profile was provided on Figure 2 and 4.

- Section 4.1.1

*In Figure 2, what is the reason for the large negative extinction value (Klett) at 190 m height? Juts provide more information to better understand the problem.*

*To my opinion, the normalization of the signal is a major potential drawback of the method, i.e., to accurately determine' to initialize the inversion. This need to be discussed in more detail, e.g., what is the influence of the selection of the normalization range?*
*What do you get when you vary it from the cloud base up to the limit (where SNR < 20)?*
*I am concerned about this, because Eq.(7) is only valid if the extinction coefficient remains constant with height, which is not the case in the clouds that you considered (with an increasing extinction coefficient profile). Usually the aerosol-free troposphere is used as boundary condition. And this is precisely the biggest problem in attempts to invert lidar signals within clouds, the lack of a boundary condition because of the complete attenuation of the laser light throughout the cloud. I am surprised that you got good results applying Eq. (7)*

**Response:**
The negative value at 190m height is related to the difficulty of accurately retrieving extinction by the slope method in the cloud base region. The values close to the cloud base (one bin below to the beginning of the cloud base) are almost always giving negative values (since in this area the true cloud extinction is not constant and is indeed rapidly increasing in a relative sense). For this reason, we can only use the slope method within the cloud, where the extinction is not changing as rapidly in a relative sense, to estimate alpha_0. We know that the slope method is only strictly valid if the extinction is constant. However we chose an altitude as deep into the cloud as the SNR allows. This helps ensure that relative extinction is constant enough so that the boundary value extinction is accurate enough to be useful in the backward Klett solution. Note: Klett, 1985 (https://doi.org/10.1364/AO.24.001638) showed that extinction profiles below $z_{o}$ can rapidly converge to the true results in optically thick conditions even with somewhat large errors in $\alpha_{o}$. This explains our results. Figure 2 is presented to show exactly this effect: it is only possible to use it higher within the cloud. This issue will be better explained in the revised version of the manuscript.

- *Page 4, line 28: Should it be ... ATB(z)=P'(z)z^2? You have P(z)z^2 ... without prime?*

**Response:**
Indeed, this will be corrected in the manuscript

- *Page 5, line 31: multiple scattering signal instead of multiple signal?*

*It should be written somewhere that you refer to single scattering + multiple scattering when you 'talk' about multiple scattering signals.*

**Response:**

Indeed, this will be corrected in the manuscript

- *Fig.4: Why do you use here the optical thickness? The blue solid line in Fig.4 should be the same as the black line in Fig.2, right? But I do not see that!*

**Response:**

The optical thickness is used in order to visualize the clear relation between the thickness of the cloud and the accuracy of the retrieved value of the cloud extinction.

The black line from Fig 2 presents only the retrieval of the extinction in accordance with the slope method, in the whole retrieval this method is only used to retrieve the value of alpha_0 and initiate the inversion. Therefore, the blue solid line form Figure 4 and black line from figure 2 are not the same.

- *Why is alpha in units of (m-1 sr) and not (m-1)? ... in Figs.2,4,6 ( in Fig.6,both axes).*
- *Fig.4 top line ... Retrieval*

**Response:**

This can be changed in the reviewed version of the manuscript.

- Fig.6 : *Why did you divide the presentation into four different optical thickness classes? I think all results could be shown in ONE figure. Furthermore, more explanations and a detailed description of the dataset would be helpful. Please state in the figure caption explicitly: What is n, what is E, what is A.*

**Response:**

The presentation was divided into four bins of optical thickness to clearly illustrate the relation between the cloud optical extinction and optical thickness and the effect on the accuracy for different values of the cloud optical thickness. It can be presented on one plot but then this dependence will be less visible.

- *From my point of view, the only (really) new aspect presented in this paper is the so-called resolution correction presented in the Appendix A. So, the question arises: Is the Appendix the best place for this important aspect? I would include it in the main paper body.*
  *To continue, it was not easy to follow the developments in the Appendix. There are many mistakes in the middle part that need to be corrected.*
  *Eq. A3: I think the whole expression should be divided by z?Eq. A4, A5 and A6: C should large.... not c?*
  *Eq. A6: Remove C/2, .... just B_i=1/2 (Bi,1+B_i,2) (without C)*
  *Page 10, Line 16: ... ratio ... instead of ... difference... , and ... illustrated....*
  *I do not understand: What is the impact of such assumptions (A8 and A9)? Please, provide more details.*
  *Eq. A8: Minus instead of plus? ...tau(z+...) – tau(z), and also ... tau(z-...) – tau(z)?*
  *Eq. A10: Middle term 1 – (...) ? of alpha' and then the term on the right there is one alpha instead*
  *Eq. A11: There is a minus 1 missing on the numerator, and also in Eq. A12*

**Response:**

The derivation of the resolution correction was moved to the appendix to increase the readability of the paper. It is an important part of the paper but we believe that the detail derivation is better placed in the

appendix. The formulas in the appendix were reviewed and corrected in the revised version of the manuscript.

---

## Referee Report (RR1)

**Reply by referee #2 to the responses of the authors**

**Response:**

Indeed we agree that the equations 2 to 5 need to be improved. In Eq 3 alpha'_m should be just alpha_m and the extra fraction in eq 4 was removed. The changes will be submitted in the new version of the manuscript.

*Reply:*

*The presentation of the equations is improved, but I still have trouble with your notation.*

*The prime symbol has not the same meaning for all variables. I would suggest to use the prime symbol (') only for alpha and P. The z' and dz' inside the integrals should be then replaced by another letter, maybe a greek letter...*

*Equation 5 looks different compared to the first version. I presume the new one is the correct one.*

**Response:**

A more detailed description of the scence used will be provided in the revised version of the manuscript.

Reply:

*How does the LWC profiles provided by DALES look like? ...   almost linear so that alpha increases according to $z \wedge 2/3$, like in Donovan et al. 2015?*

**Response:**

The negative value at 190m height is related to the difficulty of accurately retrieving extinction by the slope method in the cloud base region. The values close to the cloud base (one bin below to the beginning of the cloud base) are almost always giving negative values (since in this area the true cloud extinction is not constant and is indeed rapidly increasing in a relative sense). For this reason, we can only use the slope method within the cloud, where the extinction is not changing as rapidly in a relative sense, to estimate alpha_0. We know that the slope method is only strictly valid if the extinction is constant. However we chose an altitude as deep into the cloud as the SNR allows. This helps ensure that relative extinction is constant enough so that the boundary value extinction is accurate enough to be useful in the backward Klett solution. Note: Klett, 1985 (https://doi.org/10.1364/AO.24.001638) showed that extinction profiles below $z_{o}$ can rapidly converge to the true results in optically thick conditions even with somewhat large errors in $\alpha_{o}$. This explains our results. Figure 2 is presented to show exactly this effect: it is only possible to use it higher within the cloud. This issue will be better explained in the revised version of the manuscript.

Reply:

*It is true that the extinction does not change in the higher bins as fast as for the cloud base region, but to assume it as zero will mostly result in an underestimation of alpha_0' and thus of the retrieved extinction coefficient.*

*From section 4.4, should I understand that the 95% accuracy means that you get values 5 % smaller than the real ones? The underestimation should be cleary stated in the abstract and in the conclusions as it is a very important aspect about the retrieval scheme.*

**Response:**

The black line from Fig 2 presents only the retrieval of the extinction in accordance with the slope method, in the whole retrieval this method is only used to retrieve the value of alpha_0 and initiate the inversion. Therefore, the blue solid line form Figure 4 and black line from figure 2 are not the same.

*Reply*

*Please explain this clearly on the text... That Figure 2 shows the retrieved alpha' when one assumes Eq. (7) for the whole range, i.e. alpha=-1/2 d ln(ATB)/ dz. And not by using Eq. (5).*

*What should I get from Figure 2? That the normalization height is chosen where the slope method delivers the closest value to the true extinction profile? i.e. the blue line?*

*Finally two questions to Figure 5 and 6:*

*In Figure 5, why do you select such large ranges for tau up to 15 and for the error up to 30%?*

*In Figure 6, what is here A_alpha, the accuracy? the slope?*

---

## Author Response (AR2)

**Reply to the final comments of the Referee #2**

**Reply:**

The presentation of the equations is improved, but I still have trouble with your notation.

The prime symbol has not the same meaning for all variables. I would suggest to use the prime symbol (') only for alpha and P. The z' and dz' inside the integrals should be then replaced by another letter, maybe a greek letter...

Equation 5 looks different compared to the first version. I presume the new one is the correct one.

**Response to referee:**

Thank you for the comment and reviewing the equations again. We think that replacing z' and dz' with another letter will be confusing as it is a standard mathematical notation.

Indeed we found a mistake in Equation 5 and corrected it in the new version.

**Reply:**

How does the LWC profiles provided by DALES look like? ... almost linear so that alpha increases according to  $z \wedge 2/3$ , like in Donovan et al. 2015?

**Response to referee:**

Yes, the LWC profile increases according to  $z^{2/3}$ .

**Reply:**

It is true that the extinction does not change in the higher bins as fast as for the cloud base region, but to assume it as zero will mostly result in an underestimation of alpha\_0' and thus of the retrieved extinction coefficient.

From section 4.4, should I understand that the 95% accuracy means that you get values 5 % smaller than the real ones? The underestimation should be cleary stated in the abstract and in the conclusions as it is a very important aspect about the retrieval scheme.

**Response to referee:**

No, as indicated in the manuscript the 95% is the mean accuracy. Not in all cases is it underestimated. To illustrate this we provided Figure 4 where exact error can be seen for each profile and each height within the cloud.

**Reply**

Please explain this clearly on the text... That Figure 2 shows the retrieved alpha' when one assumes Eq. (7) for the whole range, i.e.  $alpha=-1/2 d \ln(ATB)/dz$ . And not by using Eq. (5).

What should I get from Figure 2? That the normalization height is chosen where the slope method delivers the closest value to the true extinction profile? i.e. the blue line?

**Response to referee:**

An additional note was added in the manuscript. Figure 2 together with Figure 3 are used to illustrate that the choice of the normalisation interval is crucial for the presented method. The extinction estimated from the slope method can get noisy if chosen too high above the cloud base height. As this alpha\_0 is used to initiate the inversion it is important that its value is close to the true extinction.

**Finally two questions to Figure 5 and 6:**

In Figure 5, why do you select such large ranges for tau up to 15 and for the error up to 30%? In Figure 6, what is here A\_alpha, the accuracy? the slope?

**Response to referee:**

In figure 5 we wanted to illustrate that the error of the retrieval can vary per profile and per height within the cloud. The highest value of the error obtained was 30% hence the scale. For the optical thickness we chose the value of 15 as that was the maximum value obtained for heights above 500m and we wanted to keep all the scales consistent (the scale for Radar reflectivity and Attenuated backscatter was up to 600m). In figure 6 Alpha\_a is the accuracy. We added an explicit note in the manuscript.